# Regulatory Compliance of Community-Based Conservation Organizations: Empirical Evidence from Annapurna Conservation Area, Nepal

**Nabin Baral** [1],* and **Joel T. Heinen** [2]

1   School of Environmental and Forest Sciences, University of Washington, Seattle, WA 98195, USA
2   Department of Earth and Environment, Florida International University, Miami, FL 33199, USA; heinenj@fiu.edu
*   Correspondence: nbaral@gmail.com

**Abstract:** Community-based conservation in the developing world generally puts more emphasis on voluntary commitments and compliance rather than enforcement of formal laws and regulations for the governance of protected areas. However, as with other forms of organizational management, once institutions are established, they are required to comply with all relevant, legally binding regulations. Furthermore, it is broadly assumed that compliance with established regulations is critical for good governance. In this paper, we review these matters through an empirical study of Conservation Area Management Committees' degree of compliance with regulations under Nepalese law, within the Annapurna Conservation Area—one of the best-known community-based protected areas worldwide—based on quantitative content analysis of the committees' meeting minutes from 2008 to 2012. According to the established rules, two to four women and one to five minorities serve as committee members in each instance. On average, fewer members than expected attended meetings, and the number of decisions made per meeting showed a curvilinear relationship with the number of members present as well as their demographic diversity. Of the 13 committees selected for study, only two met the legal mandate of holding six regular meetings annually within two-month intervals. In all the other cases, non-compliance was noted for one to all five years of the committees' terms. In general, compliance declined over the five-year terms, and some committees were significantly less-compliant than others. Although enforceable decisions were made within both compliant and non-compliant committees, several problems of non-compliance were identified that may affect conservation outcomes. We suggest several possible reasons for non-compliance and argue that these may be symptoms of institutional weaknesses. Organizations that fail to meet their commitments risk liability and may also lose the formal legal authority to govern. Regular monitoring is recommended to address compliance issues.

**Keywords:** community conservation; decentralized governance; empowerment; government regulations; monitoring and evaluation; participatory approaches

## 1. Introduction

### 1.1. Background

The phenomenon of "paper parks" in developing countries is widely recognized as problematic throughout the conservation literature [1–3], and it is at least partly caused by a lack of adequate governance structures and managerial oversight to enforce compliance. Furthermore, it is widely recognized that good governance of protected areas (PAs) is crucial for the conservation of biodiversity [4,5]. In an effort

to render conservation programs more effective, efficient and inclusive, and to reduce park-people conflicts common under older "fences and fines" conservation models [6], policies empowering local communities and soliciting their participation in natural resource governance have been implemented in many places; however, compliance issues often remain [7,8].

Nepal has experimented with, and implemented, community-based conservation (CBC) models for its forests and PAs in recent decades (below). Its PA system, including 20 core reserves with buffer zones around many of them, currently encompasses more than 20% of the country's land area. Buffer zones and conservations areas (CAs) together account for about 60% of the protected area estate [9] and are governed by CBC programs involving locally elected representatives.

Beginning in the late 1970s, Nepal began to loosen conservation laws by allowing some local uses of renewable resources within parks and wildlife reserves. Similar to many other developing countries, the Government of Nepal (GoN) began amending its national legislation in an effort to meet United Nations Sustainable Development Goals that recommended the inclusion of women and ethnic minorities in decision-making at all levels of governance [10–13]. While ethnic heterogeneity can be a barrier to forming CBC programs, rules that allow for inclusion are considered beneficial to sustainable management regimes [14]. Nepal's rules specified the numbers of women and minority members for forest user groups in community forests [15–17], buffer zone management committees around parks and reserves, and conservation area management committees (CAMCs) within conservation areas [18].

In the case of CAMCs, rules stipulated that meetings should occur at least six times per year (within intervals of no more than two months) and 15 members should be elected for five-year terms [18]. Members may be re-elected to foster institutional memory and under-represented minorities and women must be included as full members. There are also requirements for auditing and keeping minutes. In most instances, CAs are managed by non-governmental organizations and, in some cases, by the Department of National Parks and Wildlife Conservation (DNPWC). They correspond approximately to IUCN Category VI reserves [19]. In all cases, the DNPWC maintains the power to arrest and impose fines, but its main tasks include auditing and approving or amending regulations and projects decided upon by the CAMCs. The goal is to empower local users by delegating authority to make decisions for sustainable resource management.

To date, research in Nepal and elsewhere has explored, for example, differences in perceptions amongst different ethnic groups about demographic issues [20,21], conservation programs [22–24], natural resources uses [25] and general workings of CBC programs and other types of common properties management [14,15,26]. Much research within the conservation sector also has focused on knowledge about conservation regulations [27–29]; however, compliance with regulations (i.e., meeting representational and format rules), has received little attention to date [30,31]. Yet, compliance and enforcement are two important aspects of the effective implementation for all environmental regulations [32]. Conservation outcomes achieved by abiding with the formal rules and regulations might become more sustainable than otherwise.

Compliance theories such as rationalism and constructivism are useful in explaining non-compliance in many kinds of organizations [33–35]. According to the rationalist perspective, non-compliance is likely not deliberate, but agents may fail to comply due to limited capacity, non-recognition of non-compliance or changing social and economic contexts [36–38]. Per the constructivist perspective, compliance can result from long-term learning processes in which agents internalize socially accepted norms as a standard for compliance [39,40]. If members internalize through self-learning the view that compliance is important for meeting desired goals, then solutions can be long-lasting.

Within regulated environments, actors can be separated into compliant, reactive or resistant groups [41]. Compliant groups comply with regulations regardless of regulators' actions, while reactive groups comply based on regulators' actions, and resistant groups comply only if coerced. Intervention for reversing non-compliance may employ a range of actions, e.g., sanctions, self-reporting with verification and financial incentives or disincentives. If enforcement agencies have sufficient resources

for inspection, non-compliance is easy to detect. According to a rational perspective, agents are likely to comply when benefits outweigh costs and, according to a reactive perspective, agents may grow complacent but are likely to comply after minor rebukes or reminders. Non-compliance can also be seen as an indication of resistance, which implies hostility toward regulations or authorities. In our study region, however, no instances of resistance have been observed with CAMCs, so they appear to be compliant or reactive based on compliance theory. No empirical studies that we know of have examined the prevalence of these types of groups within CAMCs. Nonetheless, these categories can be potential indicators to monitor CAMCs and address any potential non-compliance issue.

Here we use this general framework of compliance to explore the behavior of CAMCs in the Annapurna Conservation Area (ACA), Nepal. While there can be many structural barriers to community participation, many of them are comparatively minor and can be overcome [42] and we consider the ability of CBC organizations to comply with regulations as one general indicator of institutional strength [43]. Specifically, CAMCs in full compliance are expected to function more effectively by making better and longer-lasting decisions than those in non-compliance. Since compliance may vary over time and space, we explore this issue with regard to how many members are present and the number of decisions made per meeting using data from a sample of CAMCs over five years. Our research questions are as follows: (1) To what extent are CAMCs in compliance with established rules about the frequency of and representativeness at meetings? (2) How do the number of members present, their ethnic and gender diversity and intervals between meetings influence the number of decisions made per meeting? (3) Is non-compliance consistent with reactivity or resistance? (4) How could non-compliance be addressed?

*1.2. Context and Study Area*

Nepal is a high biodiversity country [44,45] with extreme ecological variability [46] and a number of at-risk species [47–50]. In an effort to conserve its vast natural wealth, the Government of Nepal (GoN) began establishing strict PAs (e.g., National Parks, Wildlife Reserves) in the mid-1970s but, by the 1980s, a large number of park/people conflicts became apparent. As a result, legislation was amended to establish CAs as a CBC approach to management; ACA was the first CA designated and the largest PA of any kind (7629 sq km) within Nepal. The National Trust for Nature Conservation is its managing authority, but the GoN, through the DNPWC, has ultimate authority. The 57 CAMCs within ACA are instituted within the boundaries of Village Development Committees (VDCs)—the lowest administrative unit in the country until the 2015 constitution.

Per the most recent (2011) census, 87,832 people lived in 22,278 households within ACA. While many of Nepal's PAs are important for international tourism—a major sector of the economy for over five decades [51,52]—ACA is the most-visited by far [53–55]. It attracted 158,578 foreign tourists in 2017 and, under CA regulations, tourist entry fees directly support conservation and development within ACA. Unlike national parks, which are protected by Nepal's army, CAs organize and mobilize local people and rely on voluntary compliance. First implemented in the late 1980s, ACA has been considered a successful CBC model globally for over three decades [56–58].

One measure of success for CBC is whether formal rules and regulations are institutionalized within them. CAMCs make decisions about many aspects of conservation and development at local levels, but making conservation happen based on local norms and informal rules is quite different from managing a PA guided by national legislation. We assume that if CAMCs fail to comply with regulations, they cannot be considered fully successful even if they make some positive decisions and are seemingly functional, especially because non-compliant CAMCs can lose the legal authority to govern, and their decisions can be challenged in courts of law. This apparent paradox serves as an impetus to examine regulatory compliance. Specifically, we do not necessarily expect non-compliant CAMCs to be dysfunctional; rather, we expect better consensus to be reached when CAMCs are in compliance based on general predictions of common property theory [59,60]. According to CAMC regulations, compliance includes frequencies of, and intervals between, meetings per year, and minimum numbers

as well as gender and ethnic diversity of members participating in meetings, as defined by the statute [61].

## 2. Methods

### 2.1. Case Selection and Data Collection

We did fieldwork from 25 August to 31 October 2013, and in October 2016, by collecting information from 13 of the 57 CAMCs (Supplementary Materials) within ACA's Ghandruk and Lwang Unit Conservation Offices. They were chosen because: (1) ACA began in this region, so those CAMCs had more experience than others; (2) they were formed at different times, so temporal variability could be explored; (3) they had complete data (i.e., sets of minutes) available; and (4) they were more resilient than others during the Maoist insurgency—the decade-long civil war that claimed more than 17,000 human lives and changed the political system from a constitutional monarchy to a federal republic in Nepal, a context crucial to understanding the findings [62,63]. These characteristics make them "critical cases" [64]. The oldest CAMC started in 1989 while the youngest started in 1997. Originally, Ghara and Sikha VDCs formed one CAMC, but split in 2008. The CAMCs we chose had completed at least two five-year terms and elections were being held for the next term during the fieldwork.

We first met the chair and/or secretary of each CAMC to discuss the purpose of the research in 2013, and we requested minutes of all formal meetings from 2008 to 2012, i.e., the entire previous five-year term; all were provided. Minutes were in hardbound notebooks, so we made photocopies for later reference and transcribing into digital files. Generally, minutes began with a paragraph giving the date, place, chairperson's name and a list of members present (with signatures). Other attendees (e.g., invited guests) were also recorded, followed by a list of agenda items and of decisions made. Historical data such as minutes contain critical information about the way entities, organizations or systems have changed with time and do not become "outdated" in the manner that opinion surveys do [65]. We returned in 2016 to do key informant surveys [66] with 35 members including chairs and secretaries to gather more information about various regulatory aspects of CAMCs for contextual analyses.

### 2.2. Content Analysis of the Minutes

We did quantitative content analysis by extracting objective content of minutes. Because we were concerned with legal compliance, we needed to have objective measures that are clear and easy to interpret, easy to analyze through statistical tools and applicable to all CAMCs. The quantitative design served this purpose well. Individual meetings served as the unit for coding and analysis. By systematic evaluation, we converted qualitative texts into quantitative data to draw precise, useful and valid inferences amenable to statistical analyses. As volume of source materials was manageable (555 meetings with an average of three pages of minutes per meeting), we used the entire data set. Guided by the research questions, we pre-defined variables in a simple, precise manner to code for manifest content only (Section 2.3). Once specified characteristics were identified, we noted their presence and counted their frequency or measured intensity depending on the situation.

Minutes were coded on 11 variables: date; time since the previous meeting; members present; representation of men, women and minority groups; presence/absence of a chairperson, secretary and VDC representative; number of decisions made; and number of pages (Supplementary Materials). As our definitions are simple, precise and objective, any coder applying the criteria should arrive at similar results [67]. This should also hold true for personal attributes given the caste system of Hindu society in Nepal [68]; with the full names known, it is usually possible to discern gender and ethno-religious identity easily [69].

Quantitative content analysis has several benefits: it (1) is non-obstructive, (2) helps to reduce large amounts of information that would be difficult for qualitative analysis, (3) facilitates longitudinal studies using archived materials, and (4) can be used for descriptive and predictive functions [67,70].

All are applicable for comparing CAMCs across time. Coding and tabulating text elements that fall into specified categories is tedious and time-consuming [70] but we feel it was worth the effort here. While a mix of qualitative and quantitative research methods have proven useful in many applications [71] and purely qualitative methods have proven useful in many others [72], here we opted for the precision and replicability of quantitative content analysis for the above reasons. We are planning qualitative analyses for future studies derived from these data to address questions not considered here, such as the types of decisions arrived at when CAMCs are compliant versus non-compliant (see Discussion).

### 2.3. Definitions of Main Variables

*Number of meetings*: The total number of meetings organized by CAMCs per year.

*Members present*: Of 15 members per CAMC, the number present per meeting. A simple majority is required to make decisions, and there were no elected VDC chairs at the time of our research, so 7 members constituted a majority.

*Number of decisions*: The number of decisions made per meeting. Decisions concern all aspects of implementing conservation and development projects (e.g., rules for harvesting natural resources, managing tourism, etc.). We assume that some decisions have greater consequences than others, but this qualitative aspect was not captured here.

*Meeting interval*: The number of days between consecutive meetings.

*Diversity of members*: We used two measures of diversity. CAMCs have three member-categories: majority, minority and women. The majority refers to men of any ethnic group constituting a majority of the population within any CAMC. A simple ordinal scale was created by assigning 3 if all three groups, 2 if two groups, and 1 if only one group was/were represented. respectively. A more complex diversity scale was then created by using the Shannon diversity index [73,74]. For each meeting, we counted the proportion of members belonging to those three groups and a diversity index was calculated with Shannon's formula [69]. The Spearman correlation coefficient between the ordinal and ratio scales of diversity was 0.80, which was significantly positive ($p < 0.01$), and that between diversity and the total number of members present was 0.26, also significantly positive ($p < 0.05$). We thus decided to use the Shannon diversity index for further analyses.

*Committees*: We selected 13 CAMCs that differed in aspects such as the number of households and road access, potentially influencing results. To account for this heterogeneity (or to test whether the mean of the response variable is different among them), we created 12 dummies using the oldest CAMC (Ghandruk) as the baseline.

*Terms of the committees*: CAMCs are elected for five-year terms and the number of decisions might vary over time due to changes in social-ecological systems. We created four dummies to test whether we captured the temporal variability in the response variable using the first year of the term as the reference level in each case.

*Time index*: Meetings were held and recorded in sequence, so the data were temporal. To model the time series, we used a time index to control the potential influence of temporal ordering statistically. We assigned serial numbers to meetings to create a time index, which has no substantive interpretation, but is required for modeling time-series data [75].

### 2.4. Data Analysis

We used Kruskal-Wallis H tests to compare whether variables of interest (number of members present, number of meetings, number of decisions made and meeting intervals) differed among CAMCs and across terms. Given that we had count data that violated assumptions of normality, the Kruskal-Wallis H test is justified [76]. Furthermore, our data met the following assumptions required for the H test: (1) variables were measured on an ordinal or continuous scale, (2) independent variables consisted of two or more independent groups, (3) observations were unrelated, and (4) data in each group had similar distributions [76].

In our case, the response variable (the number of decisions made per meeting) was non-negative integer data. Building a linear regression model by treating count data as continuous can result in inconsistent and biased estimates [77,78]. Furthermore, our response variable was over-dispersed because its variance was 1.7 times greater than its mean. Ordinary least-square regression assumes a constant variance while Poisson regression requires that the mean equals the variance, so both approaches are unsuitable for our data. We thus chose a negative binomial regression model [77,78].

We built the negative binomial regression model by taking the number of decisions made per meeting as a response variable and the following variables as predictors: number and diversity of members present, and meeting intervals. We controlled heterogeneity among CAMCs and across five-year terms by including the relevant dummy variables. Furthermore, temporal ordering was statistically controlled by including the time index in the model. The regression equation was estimated in the following way. Let $d_i$ be the number of decisions made in a meeting $i$, and assume it to follow the distribution $d_i \sim$ Poisson $(\lambda_i)$, where $\lambda$ is the expected number of decisions made. The generalized linear regression model was specified as:

$$\begin{aligned} \log(\lambda_i) = {} & \beta_0 + \beta_1 \text{ members} + \beta_2 \text{ members}^2 + \beta_3 \text{ diversity} + \beta_4 \text{ diversity}^2 + \beta_5 \text{ meeting interval} + \\ & \beta_{6i} \text{ committees (12 dummies)} + \beta_{7i} \text{ term of committees (4 dummies)} + \beta_8 \text{ time index} + \varepsilon \end{aligned} \tag{1}$$

In bivariate scatter plots, the response variable showed nonlinear relationships with the number and diversity of members present, so we included higher-order polynomials to account for them in the model. These two variables (numbers and diversity of members present) were centered and squared before including them in the model based on Schielzeth [79].

Presenting the complete results of qualitative analysis is beyond the scope of this manuscript. Given the volume of data and in-depth qualitative analysis of the interviews conducted in 2016, we opted to submit a separate manuscript because of the word limit, but nevertheless we integrated some important qualitative results that help to explain the quantitative findings presented here. Moreover, these interviews brought an important context to interpret findings and inform the discussion.

## 3. Results

### 3.1. Number of Meetings

In total, 555 meetings were in our sample (Table 1). In 53 meetings, there was no majority present and, of 502 meetings with a majority present, 73 had meeting intervals of more than two months, making them invalid per regulations. Only 429 meetings were legally valid via fulfilling both rules. On average, 22.7% of meetings were invalid across all CAMCs, with the lowest in Lumle (3.0%) and highest in Ghachowk (50.0%). We used only legally valid meetings for further analyses (n = 429) because decisions made within them are defendable in court and enforcement actions do not arise in those cases.

**Table 1.** Name of conservation area management committees (CAMCs), their date of formation, the total number of meetings recorded and the breakdown of valid meetings in five years.

| CAMC | Year of Formation | Recorded Meetings | Valid Meetings | CAMC Term | | | | | Average |
|---|---|---|---|---|---|---|---|---|---|
| | | | | Year 1 | Year 2 | Year 3 | Year 4 | Year 5 | |
| Ghandruk | 1989 | 61 | 54 | 13 | 10 | 7 | 3 | 21 | **10.8 ± 6.8** |
| **Lumle** | 1993 | 67 | 65 | 14 | 16 | 12 | 13 | 10 | **13.0 ± 2.2** |
| Dangsing | 1994 | 35 | 21 | 8 | 2 | 3 | 4 | 4 | **4.2 ± 2.3** |
| Sikha | 1994 | 29 | 15 | 3 | 4 | 5 | 3 | 0 | **3.0 ± 1.9** |
| Narchayang | 1997 | 30 | 19 | 4 | 6 | 2 | 6 | 1 | **3.8 ± 2.3** |
| Ghara | 2008 | 37 | 26 | 10 | 5 | 4 | 5 | 2 | **5.2 ± 2.9** |
| Lwang | 1991 | 46 | 38 | 5 | 10 | 6 | 9 | 8 | **7.6 ± 2.1** |
| Rivan | 1991 | 40 | 31 | 12 | 12 | 1 | 3 | 3 | **6.2 ± 5.4** |
| Dhampus | 1992 | 39 | 29 | 10 | 1 | 5 | 5 | 8 | **5.8 ± 3.4** |
| Ghachowk | 1994 | 24 | 12 | 7 | 2 | 0 | 3 | 0 | **2.4 ± 2.9** |
| Macchapurche | 1994 | 40 | 32 | 10 | 6 | 9 | 4 | 3 | **6.4 ± 3.0** |
| Lahachowk | 1995 | 39 | 30 | 14 | 10 | 3 | 1 | 2 | **6.0 ± 5.7** |
| **Sardikhola** | 1995 | 68 | 57 | 16 | 13 | 12 | 7 | 9 | **11.4 ± 3.5** |
| | **Total** | **555** | **429** | **126 (29.4%)** | **97 (22.6%)** | **69 (16.1%)** | **66 (15.4%)** | **71 (16.5%)** | |
| | | | **Average** | **9.7 ± 3.9** | **7.5 ± 4.5** | **5.3 ± 3.7** | **5.1 ± 3.0** | **5.5 ± 5.6** | **6.6 ± 4.6** |

Note: Two CAMCs met the criterion of at least six meetings in a year while other CAMCs missed the mandate (in red fonts) for one to five years. Ghara split from Sikha in 2008.

Overall, the sampled CAMCs held 6.6 ± 4.6 meetings per year but there was variation among them ($\chi^2_{12}$ = 28.92, $p$ < 0.01). Lumle held the most meetings, followed by Sardikhola and Ghandruk. Ghachowk, Sikha and Narchayang held the fewest. At the coarse-grained analysis of the aggregate level, seven CAMCs organized at least six meetings on average annually (Table 1). However, the fine-grained analysis at the uncombined level showed that only two CAMCs (Lumle and Sardikhola) met the legal mandate for all five years (Table 1).

During their first year, CAMCs held more meetings on average, with the number of meetings decreasing in subsequent years ($\chi^2_4$ = 11.02, $p$ < 0.01). In the last three years of their terms, the average number of meetings was substantially lower than in the first two. At the coarse-grained temporal scale, CAMCs held at least six meetings per year on average in the first two years of their terms, but at the fine-grained scale, some did not meet the mandate in the first two years and the severity of non-compliance increased across years. In general, older CAMCs recorded more invalid meetings than younger CAMCs (Spearman r = 0.53, $p$ = 0.062, n = 13).

### 3.2. Number of Decisions Made Per Meeting

In total, CAMCs made 1796 decisions across their five years terms, with an average of 4.2 ± 2.7 decisions per meeting, but there was variation among them and across terms (Table 2). The Kruskal-Wallis H test showed that the mean ranks of CAMCs were significantly different ($\chi^2_{12}$ = 110.7, $p$ < 0.01): Lumle, Sardikhola and Lwang were the top three in terms of the average number of decisions made per year while Ghachowk, Dhampus and Sikha were the bottom three. The top CAMC (Lumle) made 11 times more decisions than the bottom (Ghachowk).

There was a declining trend in the average number of decisions made across five-year terms ($\chi^2_4$ = 17.2, $p$ < 0.01); in general, CAMCs made more decisions in the first year compared to the last. There was no association between the number of decisions and the duration of a CAMC's existence (Spearman r = −0.30, $p$ = 0.321, n = 13) but those holding more meetings also made more decisions per meeting on average (Spearman r = 0.89, $p$ < 0.05, n = 13).

### 3.3. Number of Participants in Meetings

Among all CAMCs sampled, an average number of 12.3 ± 0.6 men and 2.7 ± 0.6 women attended meetings. Minorities representation ranged between one and five, with an average of 2.2 ± 1.2 (Table 3). On average, 9.8 ± 2.0 members participated in meetings and participation varied significantly among CAMCs ($\chi^2_{12}$ = 110.8, $p$ < 0.01). The top three CAMCs (Narchayang, Lumle and Ghachowk) had an average of 11 members present at meetings while the bottom three (Ghara, Dhampus and Dangsing) had fewer than nine members present on average.

The average number of participants decreased steadily across terms ($\chi^2_4$ = 32.8, $p$ < 0.01) and was significantly higher in the first year compared to the last. Levels of participation in the middle three years did not differ from each other. There was no correlation between when CAMCs were formed and the average number of members attending meetings (Spearman r = 0.06, $p$ > 0.10, n = 13). The number of meetings per year was also unrelated to participation (Spearman r = 0.09, $p$ > 0.10, n = 13) and there was no difference in the number of participants if the meetings were held at shorter or longer intervals (Spearman r = 0.03, $p$ > 0.10, n = 13).

**Table 2.** Number of total decisions made by CAMCs in valid meetings and their breakdown into five years of a term. There was a great deal of variability among CAMCs and across the years in terms of average number of decisions made.

| CAMC | Year of Formation | Total Decisions | CAMC Term | | | | | Average per Meeting |
|---|---|---|---|---|---|---|---|---|
| | | | Year 1 | Year 2 | Year 3 | Year 4 | Year 5 | |
| Ghandruk | 1989 | 164 | 47 | 40 | 15 | 9 | 53 | 3.0 ± 2.2 |
| Lumle | 1993 | 407 | 67 | 99 | 76 | 99 | 66 | 6.3 ± 2.4 |
| Dangsing | 1994 | 111 | 33 | 30 | 19 | 15 | 14 | 5.3 ± 4.5 |
| Sikha | 1994 | 67 | 17 | 18 | 16 | 16 | 0 | 4.5 ± 2.4 |
| Narchayang | 1997 | 89 | 17 | 27 | 10 | 31 | 4 | 4.7 ± 1.4 |
| Ghara | 2008 | 93 | 31 | 23 | 19 | 17 | 3 | 3.6 ± 1.7 |
| Lwang | 1991 | 203 | 32 | 48 | 45 | 47 | 31 | 5.3 ± 2.6 |
| Rivan | 1991 | 146 | 47 | 65 | 7 | 12 | 15 | 4.7 ± 2.8 |
| Dhampus | 1992 | 57 | 30 | 3 | 11 | 6 | 7 | 2.0 ± 1.5 |
| Ghachowk | 1994 | 38 | 22 | 7 | 0 | 9 | 0 | 3.2 ± 2.5 |
| Macchapurche | 1994 | 95 | 31 | 29 | 21 | 11 | 3 | 3.0 ± 2.8 |
| Lahachowk | 1995 | 115 | 68 | 27 | 11 | 4 | 5 | 3.8 ± 2.1 |
| Sardikhola | 1995 | 211 | 69 | 41 | 56 | 25 | 20 | 3.7 ± 1.7 |
| | **Total** | **1796** | **511** (28.5%) | **457** (25.4%) | **306** (17.0%) | **301** (16.8%) | **221** (12.3%) | |
| | | **Average** | **39.3 ± 18.7** | **35.2 ± 25.3** | **23.5 ± 21.9** | **23.2 ± 25.6** | **17.0 ± 21.0** | **4.2 ± 2.7** |

**Table 3.** The number of men and women in the committees varies, and so does the average number of members attending the meetings among CAMCs across five years. Attendance in the first year is higher than in subsequent years. Empty cells (-) indicate no meetings held by the CAMCs in that year.

| CAMC | Year of Formation | Men | Women | Marginalized | CAMC Term | | | | | Average |
|---|---|---|---|---|---|---|---|---|---|---|
| | | | | | Year 1 | Year 2 | Year 3 | Year 4 | Year 5 | |
| Ghandruk | 1989 | 12 | 3 | 3 | 10.3 | 8.9 | 8.7 | 8.7 | 9.0 | **9.1 ± 0.7** |
| Lumle | 1993 | 12 | 3 | 3 | 12.0 | 10.5 | 10.6 | 11.2 | 11.0 | **11.1 ± 0.6** |
| Dangsing | 1994 | 12 | 3 | 3 | 9.8 | 9.5 | 8.7 | 8.5 | 8.3 | **8.9 ± 0.7** |
| Sikha | 1994 | 12 | 3 | 2 | 10.7 | 8.3 | 8.0 | 9.0 | - | **9.0 ± 1.2** |
| Narchayang | 1997 | 11 | 4 | 1 | 13.0 | 11.2 | 10.0 | 12.3 | 11.0 | **11.5 ± 1.2** |
| Ghara | 2008 | 13 | 2 | 1 | 8.9 | 7.6 | 9.3 | 7.6 | 8.0 | **8.3 ± 0.8** |
| Lwang | 1991 | 13 | 2 | 2 | 9.6 | 9.5 | 8.8 | 7.8 | 8.6 | **8.9 ± 0.7** |
| Rivan | 1991 | 13 | 2 | 1 | 11.2 | 10.7 | 11.0 | 10.0 | 10.0 | **10.6 ± 0.5** |
| Dhampus | 1992 | 13 | 2 | 5 | 10.9 | 8.0 | 8.8 | 7.6 | 7.1 | **8.5 ± 1.5** |
| Ghachowk | 1994 | 12 | 3 | 2 | 12.0 | 10.0 | - | 10.3 | - | **10.8 ± 1.1** |
| Macchapurche | 1994 | 12 | 3 | 1 | 10.5 | 9.3 | 9.0 | 11.3 | 9.3 | **9.9 ± 1.0** |
| Lahachowk | 1995 | 13 | 2 | 3 | 10.6 | 9.3 | 10.3 | 8.0 | 8.0 | **9.2 ± 1.2** |
| Sardikhola | 1995 | 12 | 3 | 2 | 10.1 | 10.1 | 9.7 | 8.7 | 8.9 | **9.5 ± 0.7** |
| | **Average** | **12.3** | **2.7** | **2.2** | **10.7 ± 1.1** | **9.4 ± 1.1** | **9.4 ± 0.9** | **9.3 ± 1.6** | **9.0 ± 1.2** | **9.8 ± 1.9** |

### 3.4. Meeting Intervals

Of the 502 meetings with a majority of members present, 73 had meeting intervals of more than two months. Meeting intervals differed substantially among CAMCs and across five-year terms (Figure 1). CAMCs held meetings at an average interval of 29 ± 17 days but the average interval varied significantly among them ($\chi^2_{12}$ = 46.9, $p$ < 0.01). Lahachowk, Sardikhola and Lumle had shortest intervals (23 ± 12, 26 ± 7 and 27 ± 3 days, respectively) while Narchayang, Dhampus and Sikha had the longest (45 ± 10, 43 ± 12, 41 ± 13 days, respectively).

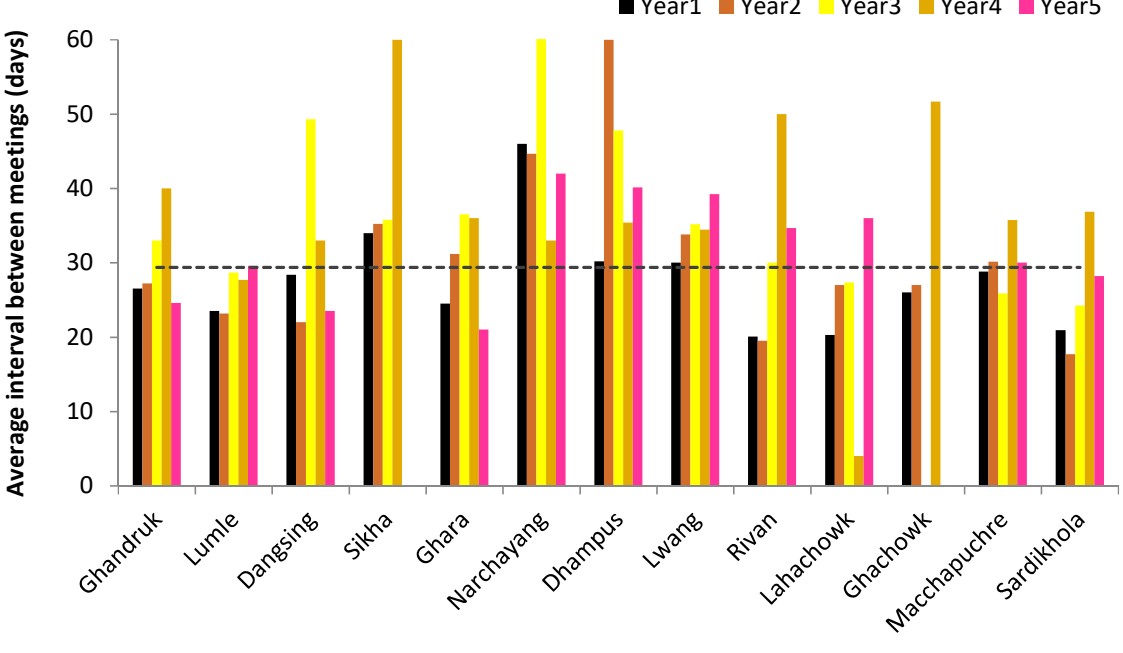

**Figure 1.** The average number of days between the two meetings (meeting intervals) show the variability among CAMCs across five years. The horizontal broken line represents the overall mean interval of 30 days.

An increasing trend in intervals between meetings was found across five-year terms ($\chi^2_4$ = 19.1, $p$ < 0.01). The average number of days between meetings in the first, second, third, fourth and fifth years of a term were, respectively, 28 ± 7 days, 31 ± 11 days, 36 ± 10 days, 37 ± 13 days and 32 ± 7 days. There was no association between CAMC age and meeting intervals (Spearman r = −0.19, $p$ > 0.10, n = 13). CAMCs holding more meetings also had shorter intervals between them (Spearman r = −0.63, $p$ < 0.05, n = 13) and those holding fewer often made fewer decisions (Spearman r = −0.65, $p$ < 0.05, n = 13). There was an increasing trend in the interval between meetings, and a decreasing trend in the number of members attending and the number of decisions made across terms (Figure 2).

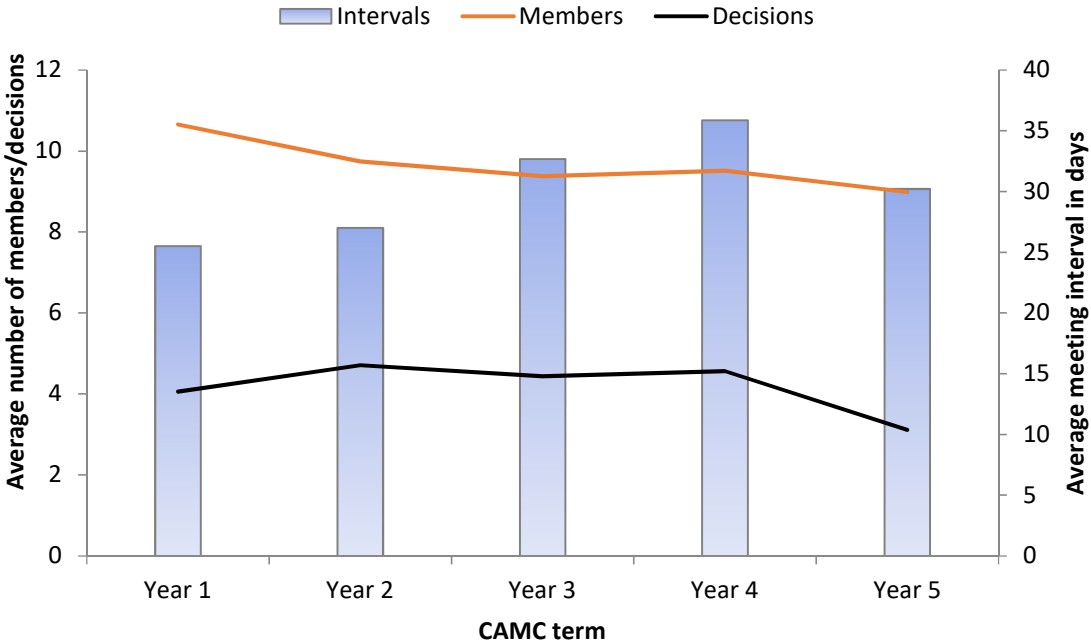

**Figure 2.** Average number of members present (Members) and decisions made (Decisions) in the meetings show a decreasing trend, and the average number of days between the two meetings (Intervals) shows an increasing trend among CAMCs across five years of their terms.

### 3.5. Predicting the Number of Decisions

The Likelihood Ratio chi-square test indicated that not all regression coefficients in the model are simultaneously zero ($\chi^2_{21}$ = 179.4, $p < 0.01$). The model with explanatory variables was significantly better than the intercept-only model to predict the number of decisions made in meetings. The dispersion parameter, alpha (=0.03), was significantly greater than zero ($\chi^2_1$ = 4.6, $p < 0.05$), suggesting that the response variable is over-dispersed, requiring a negative binomial regression tool to model such data rather than the simpler Poisson regression. The model appeared to be correctly specified as the predicted values had explanatory power in the auxiliary regression to account for the response variable's variance, while that variable's quadratic function (prediction squared) had no explanatory power (hat: z = 3.89, $p < 0.01$; hat$^2$: z = −1.47, $p = 0.142$). These results show that the generalized linear model fit the data well.

Both the linear and quadratic terms of the number of members present were significant in predicting the number of decisions made (Table 4). More decisions were made when 12 members were present compared to fewer or more than 12 (Figure 3). For each additional member, the average number of decisions made increased by 11.3% until the turning point of 12. Beyond that, the average number of decisions decreased by 4.6% per additional member.



**Table 4.** Negative binomial regression of number of decisions made on the total number of members present, diversity indices of members in the meetings, and intervals between the two meetings. The first year of CAMCs' term and Ghandruk were considered as reference levels

| Number of Decisions Made per Meeting | Coefficient | Std. Error | z | P > z | 95% Confidence Intervals | |
|---|---|---|---|---|---|---|
| Number of members (centered) | **0.113** | **0.019** | **6.07** | **0.001** | **0.077** | **0.150** |
| Number of members (centered) $^2$ | **−0.028** | **0.007** | **−4.23** | **0.001** | **−0.041** | **−0.015** |
| Members' diversity index (centered) | 0.174 | 0.164 | 1.06 | 0.288 | −0.147 | 0.496 |
| Members' diversity index (centered) $^2$ | **1.193** | **0.416** | **2.87** | **0.004** | **0.378** | **2.009** |
| Meeting intervals | 0.002 | 0.002 | 1.28 | 0.201 | −0.001 | 0.006 |
| Time index | 0.009 | 0.005 | 1.72 | 0.086 | −0.001 | 0.018 |
| CAMC term     Year 1 * | | | | | | |
| Year 2 | 0.007 | 0.095 | 0.07 | 0.942 | −0.179 | 0.192 |
| Year 3 | −0.113 | 0.138 | −0.82 | 0.413 | −0.385 | 0.158 |
| Year 4 | −0.190 | 0.175 | −1.08 | 0.278 | −0.533 | 0.153 |
| Year 5 | **−0.566** | **0.228** | **−2.48** | **0.013** | **−1.012** | **−0.119** |
| CAMC Ghandruk * | | | | | | |
| Lumle | **0.464** | **0.111** | **4.18** | **0.001** | **0.247** | **0.682** |
| Dangsing | **0.637** | **0.143** | **4.45** | **0.001** | **0.357** | **0.918** |
| Sikha | **0.439** | **0.168** | **2.62** | **0.009** | **0.110** | **0.768** |
| Narchayang | 0.229 | 0.162 | 1.42 | 0.156 | −0.088 | 0.547 |
| Ghara | **0.315** | **0.148** | **2.13** | **0.033** | **0.025** | **0.606** |
| Lwang | **0.646** | **0.129** | **5.02** | **0.001** | **0.394** | **0.899** |
| Rivan | 0.258 | 0.135 | 1.91 | 0.056 | −0.006 | 0.522 |
| Dhampus | −0.264 | 0.170 | −1.56 | 0.119 | −0.597 | 0.068 |
| Ghachowk | −0.064 | 0.198 | −0.32 | 0.748 | −0.452 | 0.325 |
| Macchapuchre | −0.132 | 0.149 | −0.88 | 0.377 | −0.425 | 0.161 |
| Lahachowk | 0.166 | 0.140 | 1.19 | 0.236 | −0.109 | 0.442 |
| Sardikhola | 0.038 | 0.122 | 0.31 | 0.756 | −0.201 | 0.276 |
| Constant | 1.074 | 0.128 | 8.39 | 0.001 | 0.823 | 1.325 |
| Alpha | 0.033 | 0.017 | | | 0.012 | 0.090 |

Note: (*) for the dummy variables representing time and committees, respectively. Gray color fonts indicate that the relationships are statistically insignificant.

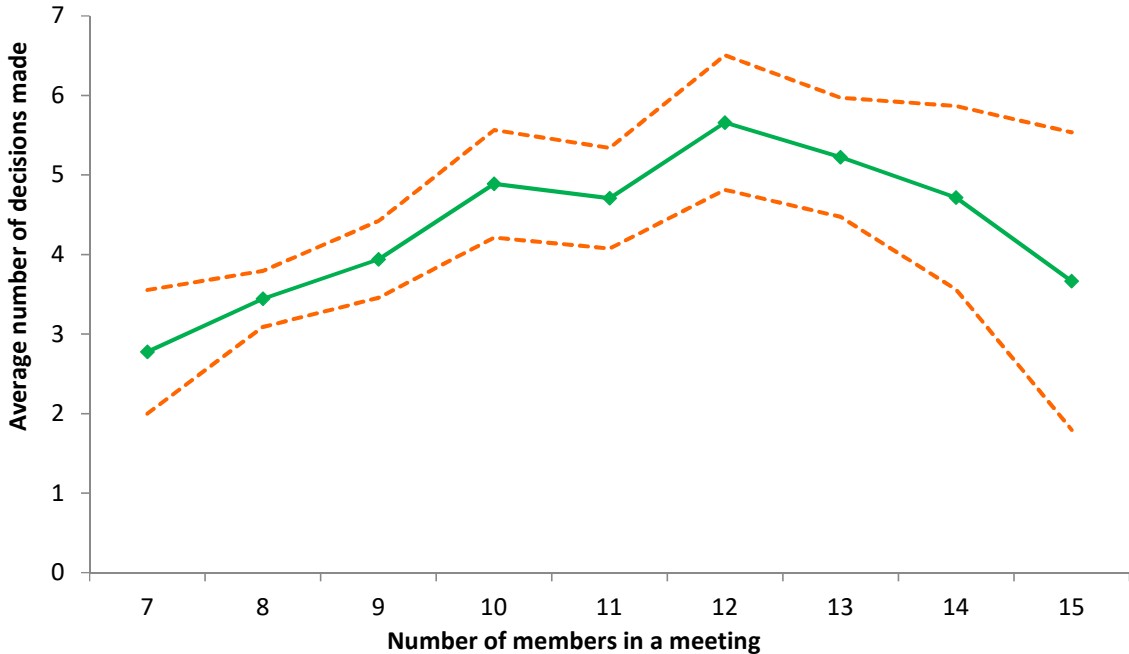

**Figure 3.** A curvilinear relationship between the number of members present in meetings and the average number of decisions made indicates that significantly more decisions are made when 12 members are present in CAMC meetings. Broken lines show confidence intervals of the estimates.

The linear term of the diversity index was not significant but the quadratic term in the regression model was. Because both linear and quadratic terms had positive coefficients, the best-fitting curve had a slope that increases with steepness, so the rate of increase in decisions was higher for extreme values of diversity. A loose interpretation would be that a one-point increase in diversity was associated with a 238.6% increase in the number of decisions made, all else held equal. At the higher and lower levels of diversity, the average number of decisions made was higher than at medium levels of diversity (Figure 4). Contrary to expectation, the time between meetings did not influence the number of decisions made at the next meeting.

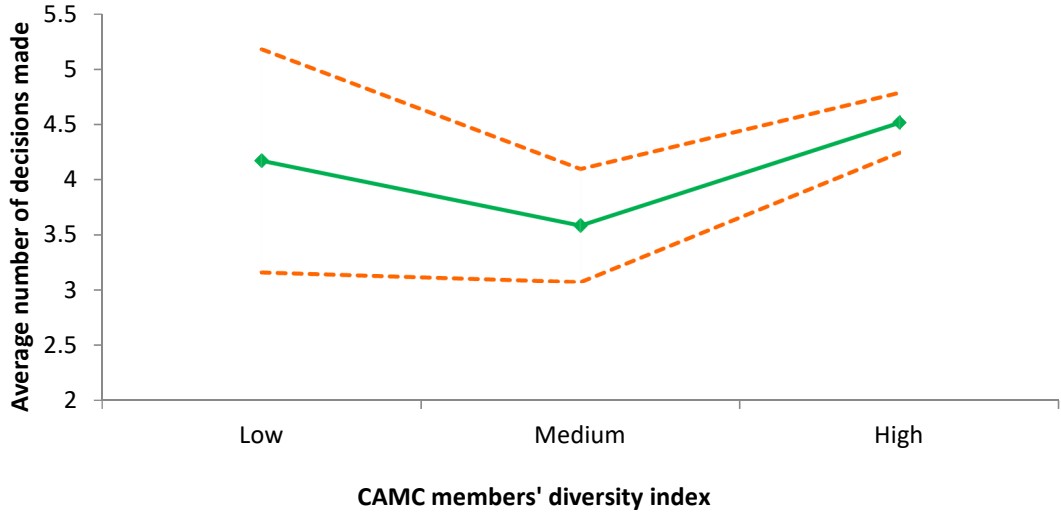

**Figure 4.** A nonlinear relationship between the diversity of members present in meetings and the average number of decisions made in CAMC meetings. Confidence intervals are shown by broken lines.

Considering variation among committees, five CAMCs: Lwang, Dangsing, Lumle, Sikha and Ghara made 64.6%, 63.7%, 46.4%, 43.9% and 31.5%, respectively, more decisions per meeting than Ghandruk. In terms of temporal variability, the first and last years of CAMC terms mattered most. The average number of decisions made in the last year was 56.6% less than in the first. Finally, in 2016 key informant interviews, CAMC members and officers were asked what they considered an optimal level of participation (n = 35). According to them, 11.5 ± 1.3 members per meeting was ideal; thus, their expectations were consistent with our findings.

## 4. Discussion

This study shows that CAMCs within ACA are in frequent non-compliance with regulations concerning the numbers and intervals of meetings, as well as the number and diversity of members in attendance, but CAMCs are expected to function while PA status remains. We found correlations between these factors and the numbers of decisions made, with the general result that non-compliance (longer intervals, fewer members present, etc.) led to fewer decisions. We also found variation within and among CAMCs, tending toward greater non-compliance as five-year terms progressed, with some CAMCs having worse overall compliance than others throughout their terms. While we have shown a good deal of non-compliance, one could argue that the findings do not necessarily mean that CAMCs are dysfunctional. A finer and longer-term analysis would be needed to address this possibility, and we are in the process of further analyses of the types of decisions made over time using primarily qualitative methods (see also [63]).

We contend, however, that non-compliance itself is a matter of some concern given that the regulations, as written, were developed using a bottom-up framework with much community involvement and full support. Much theoretical and empirical work also has shown compliance with well-accepted and agreed-upon rules to be important for long-term sustainability of socio-ecological systems [59]. As written, the regulations represent community consensus and we consider it of some consequence that elected representatives are frequently not following their own binding regulations. Further, while this has not yet happened in ACA, the cases of non-compliance could render CAMCs potentially liable to lawsuits given that only 2 of 13 cases studied here consistently met legal mandates of meeting frequency with a majority of members present. ACA is among the oldest and considered among the most successful CBC models globally [80,81], yet obvious regulatory compliance issues remain (Table 1).

Compliance issues typically arise when mandates are too demanding [39,82], but demands appear to be manageable for CAMCs given their capacity and access to monetary resources [57,61]. Key informant interviews implied that regulators typically look at total numbers of meetings—in which most CAMCs appeared to be compliant—or assume CAMCs are doing well if projects are completed and audited. Contextual factors—e.g., whether agendas are perceived as urgent to solve problems—can determine the optimal number of meetings, decisions to be made and meeting intervals. One may expect variation among CAMCs and across time in those variables, but the high variation we observed may allude to procedural or organizational inefficiencies. The two indicators examined in this paper shed some light for understanding compliance, but qualitative aspects of the decisions themselves cannot be ruled out. Non-compliance may not appear to interfere with functioning of CAMCs superficially, but its existence can be challenged legally.

Based on compliance theory ([38,83]; see Introduction), rational agents weigh costs and benefits of complying and are more likely to comply when benefits outweigh costs. Non-compliance can also be seen as an indication of resistance [33–35]. We find the constructs of rationality and resistance to be unlikely in the case of ACA. There appeared to be no additional costs to holding regular meetings, and, while our data clearly show that CAMCs are frequently non-compliant, we also found no evidence that they are resistant per UNEP [41] standards. Resistance implies some degree of hostility toward regulations and/or the authority, and we found no such evidence in any of our surveys. We suspect that resistance would have been especially evident during key informant surveys, as subjects were not

hesitant to express both positive and negative opinions when asked. We thus suspect that CAMCs behave reactively and compliance would improve if a monitoring system were in place to detect violations and give CAMCs appropriate warnings and response time.

Regular monitoring, or even a framework for it, is thus far lacking in ACA and within CA and Buffer Zone regulations nation-wide, as far as we were able to ascertain. Since these two management categories comprise about 60% of the area of the PA system of Nepal, this is a significant finding. While DNPWC officers hold approval or veto power over actions taken by CAMCs, and general annual audits of finances and project plans are required, no regular, ongoing monitoring protocols are required, nor have any been developed. Regular monitoring would also be helpful to test whether, and under what circumstances, reminders are effective or whether positive or negative incentives should be imposed per some compliance frameworks [48,84].

According to our findings, non-compliance is unlikely to be deliberate, but agents may fail to meet it due to limited capacity, inadvertence (i.e., agents failing to realize their non-compliance) or changing social and economic contexts [36–38]. Regular monitoring, and making monitoring results transparent, could again prove effective in such cases. As per the constructivist perspective [39,40], if CAMC members internalize that compliance is important through learning, then the solution can be long-lasting. It appears that changing CAMCs' preferences through social learning may thus be helpful in this situation.

The curvilinear relationship between the number of decisions made and the diversity of members present per meeting needs further explanation. The number of agenda items and the time required to deliberate each of these both factor into determining the number of decisions that can be made per meeting. We expect fewer items presented when diversity is low, but trust among members would likely be high in those cases [85], leading to consensus quickly. Conversely, we hypothesize that, when diversity is high, more agenda items are likely due to wider variance of attendee concerns. Yet, more heterogeneity may speed the deliberation process because there is no critical mass of any one interest, resulting in more decisions per meetings with high or low diversity. In medium levels of diversity, the opposing forces could be of sufficient magnitude to slow decision-making. There may not be more agenda items in a medium-diversity scenario, but deliberations could simply take longer. This hypothesis is speculative and would need testing.

Finally, we took a broad approach here and there are some limitations to consider. We looked at quantitative aspects of regular meetings while perspectives gained from qualitative data could be insightful, such as the types of decisions that tend to be made when CAMCs are compliant versus non-compliance, earlier or later in their tenure. Given the design, we were unable to consider reasons for non-compliance directly; in the ACA region, they could include anything from rainy weather during monsoon to women CAMC members' reluctance to walk alone to attend meetings, since most transport is by foot over unpaved trails. Any differences in decisions made, in terms of their influence on communities, could be further explored in more targeted qualitative contextual research over time.

## 5. Conclusions

The CAMC approach to CBC has been considered successful in ACA and other CAs in Nepal. Our study, however, shows that CAMCs are also frequently in non-compliance with regard to published regulations at finer scales, and that the number of decisions made is affected by both overall attendance and by the gender and ethnic diversity of attendees. Non-compliance with regulations could also have negative impacts on the credibility and legitimacy of CAMCs over time, the foundation on which the whole edifice of CBC stands for CAs in Nepal.

These findings have implications for environmental politics and governance. ACA is considered globally as a successful model and an important case study in CBC, but it has not fully institutionalized its own regulations over its several decades of operation. If regulatory compliance is an issue in this case [63], then it might not bode well for many other cases, and especially those that are newer and/or less resilient for other social, political or economic reasons. The hypotheses generated in this study

can be further tested in other CBC approaches in Nepal and elsewhere, such as with buffer zone management around PAs and community forest systems now in place in many countries. The common assumption is that an emphasis on voluntary compliance (as opposed to strict top-down enforcement) should lead to success of CBC programs. Our results indicate that people are meeting and making some decisions, but the organization overall is frequently not in compliance with its own stated formal regulations.

Our results imply that non-compliance in this case is most likely to be inadvertent, and we infer that CAMC members would be reactive to criticism. In such a scenario, warnings in the form of gentle reminders may be all that is needed, as opposed to more punishing (dis)incentives [84]. This suggests the need for regular monitoring and raising member awareness to address compliance issues arising in this widely referenced CBC model. Future study questions that we encourage to be considered in a multitude of CBC examples include whether bi-monthly meetings make sense, or if longer or shorter intervals depending on committee terms or season of the year would be appropriate. Given that our results showed that CAMCs with 12 members appeared to be optimal for making decisions, and key informants corroborated that finding, should the mandated number of 15 members per CAMC be changed? Could it be lowered, or variable? How many minority ethnic groups and/or women members, on average, would be optimal for decision making? What are the main barriers for compliance? These are but a few of the emerging questions to consider to further our understanding of this important conservation model, which is relevant for many protected and conserved areas in Nepal and around the world.

**Supplementary Materials:** The supplementary materials are available online at http://www.mdpi.com/2071-1050/12/22/9420/s1.

**Author Contributions:** N.B. conceptualized and designed research; administered the project; collected, organized and analyzed data; and wrote the paper and addressed the comments. J.T.H. helped in research conceptualization and design; reviewed the literature; and wrote the paper and responded to the reviews. Both authors have read and agreed to the published version of the manuscript.

**Funding:** The Rufford Small Grants, UK (Grant # 14034-B) provided financial help for conducting the fieldwork.

**Acknowledgments:** Rishi Ram Subedi and Bishnu Paudel helped in the field. Meghan Foard, Lothar Bildat, M. J. Arul and Reuven Shapira provided useful references and guided our search for the explanation of diversity results.

**Conflicts of Interest:** The authors declare no conflict of interest.

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
