# Peer review of "Regulatory Compliance of Community-Based Conservation Organizations: Empirical Evidence from Annapurna Conservation Area, Nepal"

_sustainability, doi:10.3390/su12229420_

Round 1

Reviewer 1 Report

Thank you for the opportunity to read your manuscript. The ACA and its associated CAMCs provide a meaningful case to examine PA-associated conservation organizations more broadly, in terms of metrics of compliance. The paper has merit for publication, however, requires more positioning within its contribution to science and theory first.

This is a very interesting context and adequate study design and analysis, but it lacks ties to theory that promote it as a contribution to science. For example, a richer positioning of CAMCs within the framework for groups (compliant, reactive, or resistant - lines 75-82) would make a much more compelling argument for this study. What studies have examined the prevalence of these types of groups within CAMCs or other PA-related community organizations? What do your findings in the ACA suggest about the utility of these categories and potential indicators within them? Would you suggest that the indicators you examined in the quantitative content analysis are sufficient for understanding compliance? Are any more pertinent than others? What are the relationships between these metrics of compliance and practical functioning of CAMCs and their PAs/CAs? The finding of decreasing compliance with time is really important, as is the compliance-decisions relationship, and these two could be highlighted earlier to preview the themes and act as research questions.

There is ample opportunity for ties to theory for this work and I encourage the authors to consider them. Otherwise, it reads as a good technical report but not a scientific journal article.

Related, the abstract has too much emphasis on the methods/results. It doesn't address the big questions of Why? and So what? I suggest condensing the findings (most details here are fine-detail and should be left for the narrative), starting with a compelling question of why you looked at regulatory compliance, and inserting two sentences - the major implication for theory/science and the major implication for the setting/community-based conservation organizations.

I agree that the Maoist insurgency is a crucial context (line 128). A sentence or two should be added here for readers unfamiliar with this part of Nepali history.

Please order the rows in the table by the most salient variable - year of formation, recorded or valid meetings. This would allow for better assertation of patterns across the data either by the age of the CAMC or its activity.

The discussion in general is very well written and interesting. There is a lot of new terminology and jargon introduced, however, which warrants another examination. For example, many compliance theories (line 107) are suggested but not detailed. These leave the reader hanging and thus don't contribute to the narrative. To further emphasize the point above, these could be explored in more detail earlier for the theoretical link.

These data were collected many years ago now. It would be useful to have an update in the discussion about the current status of these CAMCs - are they still functioning, have they dissolved, etc.?

The manuscript would benefit from an additional thorough edit for brevity, repetition, and some grammatical issues.

Author Response

Comments and Suggestions for Authors

Thank you for the opportunity to read your manuscript. The ACA and its associated CAMCs provide a meaningful case to examine PA-associated conservation organizations more broadly, in terms of metrics of compliance. The paper has merit for publication, however, requires more positioning within its contribution to science and theory first.

This is a very interesting context and adequate study design and analysis, but it lacks ties to theory that promote it as a contribution to science. For example, a richer positioning of CAMCs within the framework for groups (compliant, reactive, or resistant - lines 75-82) would make a much more compelling argument for this study. What studies have examined the prevalence of these types of groups within CAMCs or other PA-related community organizations? What do your findings in the ACA suggest about the utility of these categories and potential indicators within them? Would you suggest that the indicators you examined in the quantitative content analysis are sufficient for understanding compliance? Are any more pertinent than others? What are the relationships between these metrics of compliance and practical functioning of CAMCs and their PAs/CAs? The finding of decreasing compliance with time is really important, as is the compliance-decisions relationship, and these two could be highlighted earlier to preview the themes and act as research questions.

There is ample opportunity for ties to theory for this work and I encourage the authors to consider them. Otherwise, it reads as a good technical report but not a scientific journal article.

Authors’ Response: Thank you for your constructive comments. We have thoroughly revised the Introduction to link theory with a case (Manuscript pages 4 and 5), and parts of the Discussion (Manuscript pages 19 and 20) to follow up on issue. Based on our experience, no instances of resistance have been observed with CAMCs, so they appear to be either compliant or reactive based on compliance theory. No empirical studies that we know of have examined the prevalence of these types of groups within CAMCs. Nonetheless these categories can be potential indicators to monitor CAMCs and address any potential noncompliance issue (see also Conclusions page 23).

Related, the abstract has too much emphasis on the methods/results. It doesn't address the big questions of Why? and So what? I suggest condensing the findings (most details here are fine-detail and should be left for the narrative), starting with a compelling question of why you looked at regulatory compliance, and inserting two sentences - the major implication for theory/science and the major implication for the setting/community-based conservation organizations.

Authors’ Response: Thank you for the suggestion. We revised the abstract as per your suggestion. Please see the revised version.

I agree that the Maoist insurgency is a crucial context (line 128). A sentence or two should be added here for readers unfamiliar with this part of Nepali history.

Authors’ Response: We added the following sentences in the revised manuscript to provide a context for the Maoist insurgency and to make the readers familiar about it: “… the decade long civil war that claimed more than 17,000 human lives and changed a political system from constitutional monarchy to federal republic in Nepal….” We hope this would suffice.

Please order the rows in the table by the most salient variable - year of formation, recorded or valid meetings. This would allow for better assertation of patterns across the data either by the age of the CAMC or its activity.

Authors’ Response: After reading your comment, we felt that such ordering would be better for presentation. Previously, we organized the CAMCs according to their administrative units following the style of governing organization, the Annapurna Conservation Area Project. In the revised version, we organized the CAMCs by the year of formation. The first six CAMCs (Ghandruk to Ghara) belongs to one administrative unit and the latter seven CAMCs (Lwang to Sardikhola) belong to another administrative unit (of seven units in total). So, the temporal ordering of CAMCs is retained within the two administrative units.

The discussion in general is very well written and interesting. There is a lot of new terminology and jargon introduced, however, which warrants another examination. For example, many compliance theories (line 107) are suggested but not detailed. These leave the reader hanging and thus don't contribute to the narrative. To further emphasize the point above, these could be explored in more detail earlier for the theoretical link.

Authors’ Response: The reviewer is correct, and, after re-thinking these remarks, we re-worked the Introduction to include an broad outline of compliance theory up front (pages 4, 5), and we re-worked the Discussion to expand upon this issue from that (pages 20, 21), with a brief follow-up in Conclusions (page 23).

These data were collected many years ago now. It would be useful to have an update in the discussion about the current status of these CAMCs - are they still functioning, have they dissolved, etc.?

Authors’ Response: Thank you for bringing up this issue. Because these are historical data, they are not outdated. We revised the discussion to make it clear that CAMCs are perpetual entities as per the present conservation regulations. The CAMCs are local organizations within a community-based protected area established by a legislation. After our fieldwork, new members were elected to the CAMCs for the next five years. The CAMCs are still functioning and expected to function in the future too. As long as the status of protected area remains, the CAMCs are expected to function within the present legal framework.

The manuscript would benefit from an additional thorough edit for brevity, repetition, and some grammatical issues.

Authors’ Response: Thank you for your editorial remark. We thoroughly checked the manuscript for grammatical issues, repetition and brevity and revised the text wherever relevant.

Reviewer 2 Report

The introduction should be developed to clarify the importance of this particular study and why it is designed the way it is.

The research questions should be more integrated to the problem formulation.

Of specific interest is why the chosen methodology give the best answers to the questions.

The reasoning around the choice of quantitative study instead of qualitative needs to be developed. - could it have been personal reasons why some of the activities survived? Quality of the leaderships? Management structures? Relationship between the committee and the staff? Power struggling?

It seems to be new perspectives in the conclusions - these should be presented earlier in the text. And better linked to the problem formulation.

Author Response

Comments and Suggestions for Authors

The introduction should be developed to clarify the importance of this particular study and why it is designed the way it is.

Authors’ Response: The Introduction was completely re-worked with these comments in mind, the theoretical framework and the importance of this study are now (we feel) much more obvious than the previous draft.

The research questions should be more integrated to the problem formulation.

Authors’ Response: Please see above. We have directly articulated the research questions at the end of the Introduction (pages 3, 4) following from the re-write of other parts above it (pages 5, 6)

Of specific interest is why the chosen methodology give the best answers to the questions.

Authors’ Response: We have revised the methods section to address this comment. Our primary interest was to measure the degree of compliance with regulations taking two major indicators explicitly mentioned in the regulations: the number of meetings and their intervals. Because our research questions can only be answered by the quantification of those events, we need to design a quantitative study to collect data-driven evidence. As such, quantitative methods served the purpose better to get the best possible answer to our research questions. Our quantitative research design helped to answer questions such as “how many” and “how often” the CAMCs’ formal meetings were organized that are critical indicators to assess the degree of compliance. Because we are concerned with factual information, we need to have objective measures that are clear but harder to misinterpret, easy to analyze through statistical tools, and applicable to all CAMCs.

The reasoning around the choice of quantitative study instead of qualitative needs to be developed. - could it have been personal reasons why some of the activities survived? Quality of the leaderships? Management structures? Relationship between the committee and the staff? Power struggling?

Authors’ Response: Quantitative and qualitative research methods are two different approaches, based on different paradigms and different assumptions about ontology and epistemology. The question how come some CAMCs are compliant and others are not can be answered by a qualitative design. As you suggested, both the organizational variables such as the quality of leaders and personal level variables such as the motivations and barriers for members to participate in CAMC meetings made a difference in the outcomes of CAMCs. We are preparing another manuscript to answer your questions based on qualitative research conducted.

It seems to be new perspectives in the conclusions - these should be presented earlier in the text. And better linked to the problem formulation.

Authors’ Response: As indicated above, we have followed the comments carefully and we have re-worked the Introduction and Discussion to address this issue. The theoretical compliance framework is introduced on pages 3, 4; the research questions are expanded upon on pages 5, 6; compliance is discussed in light of our findings on pages 20, 21; and concluded on page 23.

Round 2

Reviewer 1 Report

Thank you for your detailed responses to my comments, in the letter and on the manuscript. This is a highly enjoyable paper to read, and the efforts you have made have really strengthened it. I look forward to sharing it once published. Please conduct a final sentence-level edit for a few small remaining typos and language use issues.

Author Response

Authors’ Response: We are glad to know that you enjoyed reading our paper. Thank you for pointing the typos. We combed the paper thoroughly for typos and fixed them.